# Efficacy and safety of single-dose 40 mg/kg oral praziquantel in the treatment of schistosomiasis in preschool-age versus school-age children: An individual participant data meta-analysis

Piero L. Olliaro[1], Jean T. Coulibaly[2,3,4,5], Amadou Garba[6], Christine Halleux[7], Jennifer Keiser[2,3], Charles H. King[8,9], Francisca Mutapi[10,11], Eliézer K. N'Goran[4,5], Giovanna Raso[2,3], Alexandra U. Scherrer[12], José Carlos Sousa-Figueiredo[13,14], Katarina Stete[15], Jürg Utzinger[2,3], Michel T. Vaillant[16]*

1 Centre for Tropical Medicine and Global Health, Nuffield Department of Medicine, University of Oxford, Oxford, United Kingdom, 2 Swiss Tropical and Public Health Institute, Basel, Switzerland, 3 University of Basel, Basel, Switzerland, 4 Unité de Formation et de Recherche Biosciences, Université Félix Houphouët-Boigny, Abidjan, Côte d'Ivoire, 5 Centre Suisse de Recherches Scientifiques en Côte d'Ivoire, Abidjan, Côte d'Ivoire, 6 Department of Control of Neglected Tropical Diseases, World Health Organization, Geneva, Switzerland, 7 UNICEF/UNDP/World Bank/WHO Special Programme for Research and Training in Tropical Diseases (TDR), World Health Organization, Geneva, Switzerland, 8 Center for Global Health and Diseases, Case Western Reserve University, Cleveland, Ohio, United States of America, 9 Schistosomiasis Consortium for Operational Research and Evaluation, University of Georgia, Athens, Georgia, United States of America, 10 NIHR Global Health Research Unit Tackling Infections to Benefit Africa (TIBA), Ashworth Laboratories, University of Edinburgh, Edinburgh, United Kingdom, 11 Institute of Immunology and Infection Research, Centre for Immunity, Infection and Evolution, School of Biological Sciences, Ashworth Laboratories, University of Edinburgh, Edinburgh, United Kingdom, 12 Division of Infectious Diseases and Hospital Epidemiology, University Hospital Zurich, University of Zurich, Zurich, Switzerland, 13 Department of Life Sciences, Natural History Museum, Wolfson Wellcome Biomedical Laboratories, London, United Kingdom, 14 Centro de Investigação em Saúde de Angola, Hospital Provincial, Bengo, Angola, 15 Division of Infectious Diseases, Department of Medicine II, University Medical Center Freiburg, University of Freiburg, Freiburg, Germany, 16 Centre of Competences for Methodology and Statistics, Luxembourg Institute of Health, Strassen, Luxembourg

* michel.vaillant@lih.lu

**Data Availability Statement:** Data were not produced by the current Meta-Analysis and only

## Abstract

### Background

Better knowledge of the efficacy and safety of single-dose 40 mg/kg oral praziquantel in preschool-age children is required, should preventive chemotherapy programs for schistosomiasis be expanded to include this age group.

### Methodology

We analyzed individual participant-level data from 16 studies (13 single-arm or cohort studies and three randomized trials), amounting to 683 preschool-age children (aged <6 years) and 2,010 school-age children (aged 6–14 years). Children had a documented *Schistosoma mansoni* or *S. haematobium* infection, were treated with single 40 mg/kg oral praziquantel, and assessed between 21 and 60 days post-treatment. Efficacy was expressed as

the original authors could make them available on an individual basis. These data are held in the repository of the Infectious Diseases Data Observatory (IDDO.org). IDDO promotes data sharing and data re-use to generate new evidence that improves health and understanding of disease. Requests to access data can be submitted by email to dataccess@iddo.org via the Data Access Application Form available at IDDO.org/accessing-data. If eligible, requests will be reviewed by the IDDO Data Access Committee to ensure that use of data protects the interests of the participants and researchers according to the IDDO principles of data sharing (see https://www.iddo.org/data-sharing/accessing-data).

**Funding:** The Special Programme for Research and Training in Tropical Diseases (TDR) supported this work and the clinical trial reported in Olliaro et al. (2013) with core funding. The Luxembourg Institute of Health supported the analyses with core funding. The data from Zimbabwe was collected during a research project funded by the World Health Organization (FM) and Thrasher Research Fund (FM). This research was commissioned in part by the National Institute for Health Research (NIHR) Global Health Research programme (16/136/33) using UK aid from the UK Government (FM). JK and JTC are grateful to the European Research Council for financial support (ERC-2013-CoG 614739-A_HERO). CHK is supported by the Schistosomiasis Consortium for Operational Research and Evaluation (SCORE), funded by the University of Georgia Research Foundation through a grant from the Bill & Melinda Gates Foundation. The funders had no role in study design, data collection and analysis, decision to publish, or preparation of the manuscript.

**Competing interests:** The authors have declared that no competing interests exist.

arithmetic mean and individual egg reduction rate (ERR) and meta-analyzed using general linear models and mixed models. Safety was summarized using reported adverse events (AEs).

## Principal findings

Preschool-age children had significantly lower baseline *Schistosoma* egg counts and more losses to follow-up compared to school-age children. No difference in efficacy was found between preschool- and school-age children using a general linear model of individual-participant ERR with baseline log-transformed egg count as covariate and study, age, and sex as fixed variables, and a mixed model with a random effect on the study. Safety was reported in only four studies (n = 1,128 individuals); few AEs were reported in preschool-age children 4 and 24 hours post-treatment as well as at follow-up. Three severe but not serious AEs were recorded in school-age children during follow-up.

## Conclusions/significance

There is no indication that single-dose 40 mg/kg oral praziquantel would be less efficacious and less safe in preschool-age children compared to school-age children, with the caveat that only few randomized comparisons exist between the two age groups. Preventive chemotherapy might therefore be extended to preschool-age children, with proper monitoring of its efficacy and safety.

## Author summary

Schistosomiasis is a diseases caused by helminths (parasitic worms) which affects the intestinal and urogenital systems. In areas where schistosomiasis is endemic, the disease is controlled by the large scale distributing of praziquantel, primarily targeting school-age children. Younger children (preschool-age) too might be affected by schistosomiasis, but are currently not receiving praziquantel within treatment campaigns. Instead, preschool-age children are treated on a case-by-case basis because the current praziquantel formulation is not adapted to young children. Questions have also been raised as to whether the standard dose of 40 mg/kg given once is effective in preschool-age children. To answer this question, we collected individual-participant data from a series of studies in which 40 mg/kg of praziquantel had been given to children with intestinal or urinary schistosomiasis, and compared its efficacy and tolerability across age-groups. Since few direct comparisons had been made, we used statistical tools to make these comparisons. We found no evidence that treatment is less efficacious in preschool- than in school-age children and conclude that 40 mg/kg praziquantel may be given to preschool-age children in large-scale programs. When this happens, efficacy and tolerability will have to be closely monitored.

## Introduction

The global schistosomiasis control strategy relies upon preventive chemotherapy with praziquantel, primarily targeting school-age children. In moderate- and high-risk communities, treatment is also extended to adults [1]. Of note, preschool-age children contribute a

considerable fraction of the total burden of schistosomiasis [2–4]. The current World Health Organization (WHO) guidelines are that preschool-age children should be treated on a case-by-case basis upon diagnosis of infection due to a lack of an age-appropriate formulation of praziquantel [5]. WHO is considering the inclusion of preschool-age children in preventive chemotherapy with praziquantel, should an appropriate formulation of praziquantel become available [5–7]. The reason is that the current formulation (large, bitter tasting 600 mg tablets), although often crushed and dissolved in practice, is unsuited for use in young children, and hence, efforts are underway to develop an orally dispersible tablet formulation for young children [8, 9]. Evidence of efficacy and safety of praziquantel in preschool-age children is limited [5], and it is unclear whether they should receive the same dose (i.e., oral administration at a single dose of 40 mg/kg body weight) as their school-age counterparts, adolescents, and adults [10].

To address this issue, we analyzed data from clinical trials and epidemiologic studies that enrolled preschool- and school-age children who were treated with praziquantel at a single 40 mg/kg oral dose, for which data were available at the individual participant level. This information is important both for treatment recommendations and for adapting the strength of praziquantel to be used in pediatric formulation.

## Methods

### Ethics statement

This is a secondary analysis of published work. Ethical approval and written (or oral) informed consent have been reported in the original papers [5, 11–30].

### Datasets

Based on a scoping paper [31], the WHO Special Programme for Research and Training in Tropical Diseases (TDR) and the Department of Control of Neglected Tropical Diseases at WHO contacted investigators for the availability of suitable patient datasets from studies that enrolled preschool-age children. The investigators of 23 clinical studies [5, 11–30] agreed to share data with the specific purpose of pooled analyses aimed at answering the PICO (population, intervention, control, and outcome) question below. The datasets were curated in order to allow for the pooled analysis. One article [18] contained two different studies, which were analyzed separately. Subsequently, studies were further assessed as to their eligibility for inclusion in the analysis (whole study or subset of participants). In this dataset, studies are identified by the name of the main data contributor and the year the study was conducted.

### PICO question

In preschool-age children, is praziquantel given at 40 mg/kg body weight in a single oral dose as efficacious as it is in school-age children in reducing *Schistosoma* infection (measured as egg counts in stool or urine)?

### Study and patient inclusion criteria

The following inclusion criteria were employed: (i) treatment with praziquantel 40 mg/kg body weight; (ii) participant's age 0–14 years; (iii) confirmed infection with *Schistosoma mansoni*, *S. haematobium*, or *S. japonicum*, as determined by the presence of eggs in stool or urine; and (iv) treatment outcome assessed at follow-up visit between 21 and 60 days post-treatment.

### Assessment of methodological quality

Key characteristics of studies were extracted from the published articles. The methodologic quality was assessed through the Cochrane Collaboration's [32] risk of bias table, including items such as random sequence generation (selection bias), allocation concealment (selection bias), blinding of participants and personnel (performance bias), blinding of outcome assessment (detection bias), incomplete outcome data (attrition bias), and selective reporting (reporting bias). We also prepared funnel plots to check for publication bias, stratified by *Schistosoma* species.

### Assessment of heterogeneity

We investigated heterogeneity by examining the forest plots, and carried out sensitivity analyses by calculating the pooled mean difference of individual egg reduction rate (ERR) between preschool- and school-age children.

### Assessment of reporting bias

We compared studies included in this analysis with those identified by a prior scoping review [31]. However, only studies for which individual participant-level data were made available could be included in the present analysis.

### Statistical methods

**Summarizing infection intensity.** The arithmetic mean (AM) eggs per gram of stool (EPG) was calculated at pre- and post-treatment for *S. mansoni* by multiplying the mean individual fecal egg counts (FECs) obtained by a single, duplicate, or quadruplicate Kato-Katz thick smears (41.7 mg) by a factor of 24 [33, 34]. For *S. haematobium*, egg counts were presented as eggs per 10 ml of urine [34].

**Measuring efficacy.** Drug efficacy was expressed as AM egg reduction rate (ERR) (the difference in AM egg counts between pre- and post-treatment assessments), cure rate (CR, proportion of cases with zero egg counts post-treatment), and mean of individual ERR, with 95% confidence intervals (CIs). Individual ERRs were calculated as the ratio of the difference between the pre- and post-treatment EPG or eggs per 10 ml urine, multiplied by 100. In this analysis, no change or increase in egg counts post-treatment indicates an ERR = 0 (no reduction). CIs were determined using a bootstrap resampling method (with replacement) over 1,000 replicates. This methodology has been described in greater detail elsewhere [35]. According to WHO guidelines, the reference target efficacy for AM-ERR is ≥95% [36].

The distribution of the individual ERRs was plotted by using histograms of the frequencies and scatterplot of the cumulative frequencies as 'centiles plots'. Forest plots were utilized to visualize mean individual ERRs by age categories. Results are presented separately by *Schistosoma* species.

**Statistical analyses.** Modeling of baseline log-transformed egg counts was performed with age categories and country as fixed factors in a general linear model, and with country as a random factor in a mixed model. Modeling of the individual ERRs was carried out through a general linear model and mixed models. Separate models were fitted for *S. mansoni* and *S. haematobium*.

In the general linear model, the level of infection at baseline was included as covariate (log-transformed baseline EPG for *S. mansoni* and eggs per 10 ml of urine for *S. haematobium*). Fixed variables were country, participant's sex, and three age categories: (i) 0 to <6 years; (ii) 6 to <10 years; and (iii) 10–14 years to more accurately reflect the age-range of the included

studies (see below), or preschool-age (0 to <6 years) versus school-age children (6–14 years). Mixed models were further fitted with a random effect on the country. Sensitivity analyses were conducted with the same models by removing the baseline log-transformed egg counts.

Pairwise differences (with a Tukey adjustment) in least square means (LSM) were performed for each of the age groups. This post-hoc comparison was allowed by the implicit network of possible preschool- and school-age children comparisons across all studies (S1 Fig) [37, 38].

All tests were two-tailed and a p-value of 5% was deemed statistically significant. Calculations and analyses were performed by using Revman version 5.3.5 (The Nordic Cochrane Centre; Copenhagen, Denmark) (The Cochrane Collaboration, 2014) and SAS system version 9.3 (SAS Institute; Cary, United States of America).

Safety was assessed using reported adverse events (AEs), classified as mild, moderate, or severe. We extrapolated the number of patients exposed and assessed for safety at 4 and 24 hours post-treatment and at the end of follow-up and calculated the frequency of those with at least one AE. We also report the total number and type of AEs for each age-category and by severity.

The PRISMA guidelines were used and followed for reporting the current work. The PRISMA checklist is attached as supplementary material.

## Results

Data were available from 23 studies with children treated either for *S. mansoni*, *S. haematobium*, or *S. japonicum* infection with single 40 mg/kg oral praziquantel (Table 1 and S1 Table, including diagnostic approach used). The study flowchart (overall and by age-group; preschool- versus school-age children) is presented in Fig 1. Details by study and age-group (preschool- versus school-age) are summarized in Table 2.

### Exclusions

A total of seven studies and four study arms were excluded for the following reasons. First, six studies were excluded as a whole, as they did not meet one or more of the inclusion criteria (two studies because the praziquantel dose was not 40 mg/kg [20, 28], and four studies because the duration of follow-up was >60 days) [5, 18, 29]. Second, we excluded study arms that were outside the set criteria, namely those who received a praziquantel dose higher or lower than 40 mg/kg [13, 14, 22], or were outside the 0–14 years age range [22]. Third, we excluded participants with *S. japonicum* infection because only one preschool-child was enrolled in Xu et al. [30] out of six participants, and none in Olliaro et al. [22] (Table 1).

The remaining 16 studies and study groups enrolled a total of 4,484 (63%) children who were treated with single 40 mg/kg oral praziquantel: preschool-age children (n = 1,422; 32%) and school-age children (n = 3,062; 68%). Of note, five studies [11, 15, 22, 27] did not enroll preschool-age children. Sousa-Figueiredo et al. [25] enrolled both preschool- and school-age children aged 6–10 years, and Olliaro et al. [22] included only school-age children aged 10–14 years (Table 1). Overall 75% of the treated children (n = 2,675) were followed up and had a measurable outcome 21–60 days post-treatment; 665 preschool-age children and 2,010 school-age children. More losses to follow-up occurred among the preschool-age children compared to their older counterparts (33% versus 21%, p <0.001).

Out of the 16 studies included in the analyses, three were randomized controlled trials (RCT) and 13 were single-arm intervention or cohort studies. The RCTs were at low risk of selection bias with computer-generated block randomization, adequate allocation concealment, and blinding of either participants, personnel, or outcome assessment. The single-arm

**Table 1. Characteristics of available datasets and numbers of participants enrolled and included in the meta-analysis (0% analysed indicate excluded studies with 0% analysed).**

| Author, year of study [Ref.] | Country | Total enrolled | Follow-up duration (days) | Dose (mg/kg) | Species | Enrolled (age 0 to < 6 years) | Enrolled (age 6 to <10 years) | Enrolled (age 10 to 14 years) | % analysed of enrolled | Reason for exclusion |
|---|---|---|---|---|---|---|---|---|---|---|
| Coulibaly, 2011 [12] | Côte d'Ivoire | 53 | 21 | 40 | S. mansoni S. haematobium | 53 | | | 100% | |
| Coulibaly, 2017 [13] | Côte d'Ivoire | 84 | 21 | 20 40 60 | S. mansoni | 40 | 22 | 21 | 96% | Dose = 20 & 60 mg/kg and placebo excluded |
| Coulibaly, 2018 [14] | Côte d'Ivoire | 346 | 21 | 20 40 60 | S. haematobium | 170 | 112 | 56 | 98% | Dose = 20 & 60 mg/kg and placebo excluded |
| Garba, 2007 [15] | Niger | 659 | 42 | 40 | S. mansoni S. haematobium | | 370 | 289 | 83% | |
| Tohon, 2008 [39] | Niger | 877 | 21 | 40 | S. haematobium | | 209 | 211 | 83% | |
| Garba, 2013 [16] | Niger | 243 | 42 | 40 | S. mansoni S. haematobium | 243 | | | 95% | |
| N'Goran, 2000 [20] | Côte d'Ivoire | 354 | 52 | 80 | S. haematobium | 5 | 174 | 129 | 0% | Dose = 80 mg/kg |
| Landouré, 2006 [18] | Mali | 415 | 365 | 40 | S. mansoni S. haematobium | | 413 | 553 | 0% | Follow-up >60 days |
| Landouré, 2009 [18] | Mali | 415 | 182 | 40 | S. mansoni S. haematobium | 409 | | | 0% | Follow-up >60 days |
| Garba, 1996 [17] | Niger | 560 | 60 | 40 | S. haematobium | 77 | 86 | 76 | 63% | |
| Mutapi, 2010 [19] | Zimbabwe | 535 | 42 | 40 | S. mansoni S. haematobium | 132 | 351 | 83 | 30% | |
| Campagne, 2008 [11] | Niger | 114 | 30 | 40 | S. mansoni S. haematobium | 1 | 37 | 66 | 87% | |
| Olds, 1999 [21] | Kenya | 415 | 45 | 40 | S. haematobium | 2 | 49 | 67 | 100% | |
| Olliaro, 2007 [22] | Brazil Mauritania Philippines | 856 | 21 | 40 60 | S. mansoni S. haematobium S.japonicum | | | 534 | 36% | S. japonicum (Philippines, no preschool-age children) and dose = 60 mg/kg excluded |
| Raso, 2004 [23] | Mali | 545 | 42 | 40 | S. mansoni | 4 | 12 | 22 | 100% | |
| Sacko, 2009 [5] | Mali | 415 | 180 | 40 | S. mansoni S. haematobium | 415 | | | 0% | Follow-up >60 days |
| Scherrer, 2007 [24] | Côte d'Ivoire | 49 | 20 | 40 | S. mansoni | 6 | 22 | 21 | 100% | |
| Sousa-Figueiredo, 2012 [25] | Uganda | 880 | 21 | 6040 | S. mansoni | 693 | 187 | | 35% | |
| Stete, 2010 [26] | Côte d'Ivoire | 545 | 21 | 40 | S. haematobium | 1 | 20 | 56 | 100% | |
| Utzinger, 1997 [27] | Côte d'Ivoire | 209 | 28 | 40 | S. mansoni | | 27 | 56 | 100% | |

*(Continued)*

**Table 1.** (Continued)

| Author, year of study [Ref.] | Country | Total enrolled | Follow-up duration (days) | Dose (mg/kg) | Species | Enrolled (age 0 to < 6 years) | Enrolled (age 6 to <10 years) | Enrolled (age 10 to 14 years) | % analysed of enrolled | Reason for exclusion |
|---|---|---|---|---|---|---|---|---|---|---|
| **Utzinger, 1998 [28]** | Côte d'Ivoire | 253 | 28 | 60 | *S. mansoni* | | 129 | 124 | 0% | Dose = 60 mg/kg |
| **Wami, 2014 [29]** | Zimbabwe | 303 | 84 | 40 | *S. mansoni S. haematobium* | 109 | 148 | 46 | 0% | Follow-up >60 days |
| **Xu, 2007 [30]** | China | 880 | 90 | 40 | *S. japonicum* | 1 | | 5 | 0% | *S. japonicum* only one preschool-age child |
| **TOTAL** | | **10,005** | | | | **2,361** | **2,368** | **2,415** | | |

intervention or cohort studies were at unclear risk of bias as there was no randomization, no allocation concealment, and no blinding. Furthermore, no study mentioned if sampling was stratified for preschool- and school-age children. Regarding incomplete outcome data and selective reporting items of the risk of bias tables, attrition rate was generally low (Figs 2 and 3). The funnel plots showed extensive publications bias (Figs 4 and 5). However, for both *S. mansoni* and *S. haematobium*, half of the studies could not be plotted because they were non-comparative, hence a mean difference and a standard error of the mean between preschool- and school-age children could not be calculated (see also Figs 2 and 3).

Of the 2,010 evaluable school-age children, 988 were aged 6 to <10 years (382 (56%) pre-senting with *S. mansoni* and 301 (44%) with *S. haematobium* infections), and 1,022 were aged

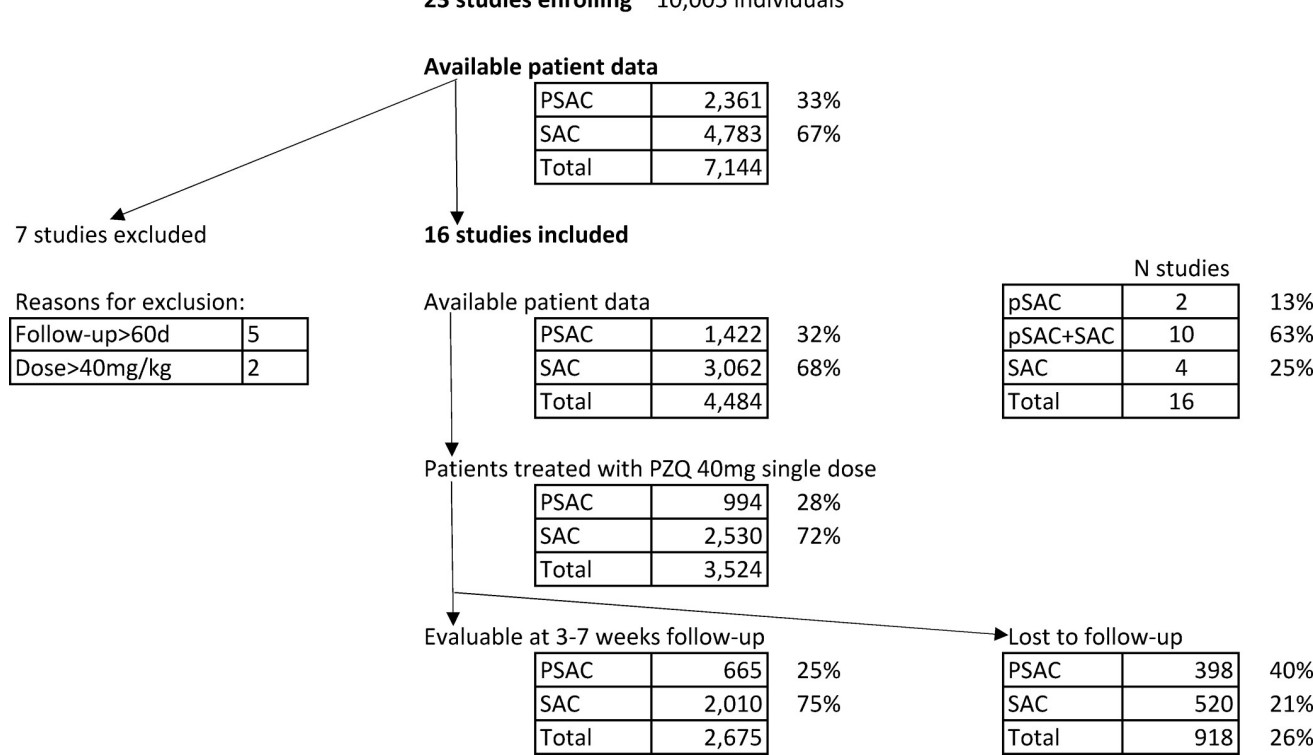

**Fig 1. Study flowchart (PSAC: preschool-age children; SAC: school-age children).**

**Table 2. Number of subjects in the 16 studies enrolled and analysed.**

| Author, year [Ref.] | Enrolled PSAC | Enrolled SAC | Enrolled PSAC+SAC | Evaluable PSAC | Evaluable SAC | Evaluable PSAC+SAC | With follow-up PSAC | With follow-up SAC | With follow-up PSAC+SAC |
|---|---|---|---|---|---|---|---|---|---|
| Coulibaly 2011 [12] | 53 | 0 | 53 | 53 | | 53 | 53 | | 53 |
| Coulibaly 2017 [13] | 40 | 43 | 83 | 38 | 42 | 80 | 38 | 42 | 80 |
| Coulibaly 2018 [14] | 170 | 168 | 338 | 37 | 37 | 74 | 37 | 37 | 74 |
| Garba 2007 [15] | 0 | 659 | 659 | | 659 | 659 | | 549 | 549 |
| Garba 2009 [17] | 0 | 420 | 420 | | 360 | 360 | | 347 | 347 |
| Garba 2013 [16] | 243 | 0 | 243 | 243 | | 243 | 231 | | 231 |
| Garba 1996 [17] | 77 | 162 | 239 | 114 | 228 | 342 | 61 | 160 | 221 |
| Mutapi 2010 [19] | 132 | 434 | 566 | 101 | 434 | 535 | 21 | 149 | 170 |
| Campagne, 2008 [11] | 1 | 103 | 104 | | 100 | 100 | | 90 | 90 |
| Olds, 1999 [21] | 2 | 116 | 118 | 2 | 116 | 118 | 2 | 116 | 118 |
| Olliaro 2007 [22] | 0 | 534 | 534 | | 190 | 190 | | 190 | 190 |
| Raso 2004 [23] | 4 | 34 | 38 | 4 | 34 | 38 | 4 | 34 | 38 |
| Scherrer 2007 [24] | 6 | 43 | 49 | 6 | 43 | 49 | 6 | 43 | 49 |
| Sousa-Figueiredo 2012 [25] | 693 | 187 | 880 | 395 | 128 | 523 | 211 | 94 | 305 |
| Stete 2010 [26] | 1 | 76 | 77 | 1 | 76 | 77 | 1 | 76 | 77 |
| Utzinger 1997 [27] | 0 | 83 | 83 | | 83 | 83 | | 83 | 83 |
| TOTAL | 1,422 | 3,062 | 4,484 | 994 | 2,530 | 3,524 | 665 | 2,010 | 2,675 |

10–14 years (667 (33%) *S. mansoni* and 1,343 (67%) *S. haematobium* infections) (Table 3). Intensity of infection at baseline and treatment outcomes expressed as ERRs calculated as AM as well as CRs are presented in Table 4, stratified by *Schistosoma* species for the three age groups (details by study in Supplementary Tables 2 and 3, stratified by *Schistosoma* species). The baseline intensity of infection analyses adjusted on study and sex showed a significant difference between age groups with higher counts in the school- than the preschool-age children (Fig 6, S4 Table, S4 Table, S5 Table, and S6 Table). A significant difference in baseline egg counts between boys and girls was found for *S. haematobium* but not for *S. mansoni* (S6 Table). The age distribution of participants by *Schistosoma* species can be found in S2 Fig.

The AM-ERRs are also presented graphically as forest plots in Figs 7 and 8 for *S. mansoni* and *S. haematobium*, respectively against the ≥95% WHO threshold for efficacy [36]. Overall, 6/13 (2/6 for *S. mansoni* and 4/7 for *S. haematobium*) of the study groups with participants in the age-group under 6 years (preschool-age) met the WHO efficacy threshold, compared to 5/

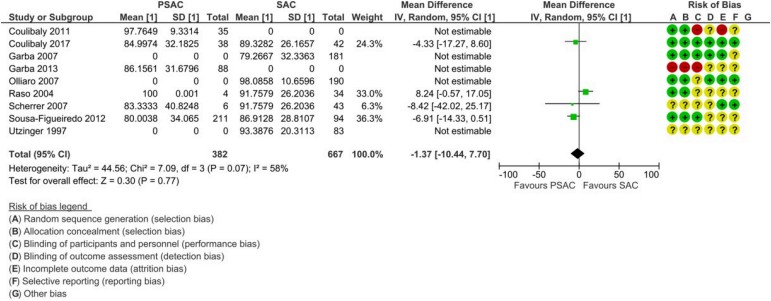

**Fig 2. *S. mansoni* studies forest plot of mean egg counts and bias table.**

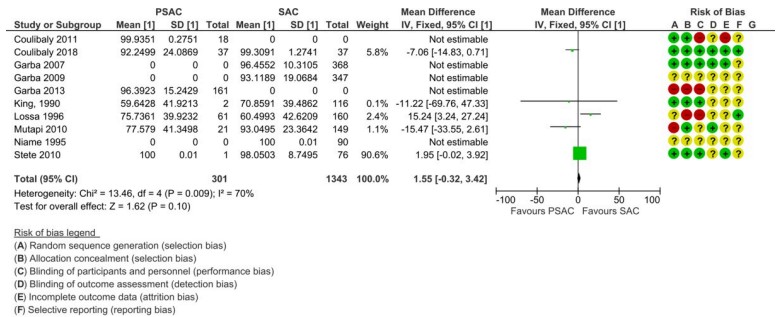

**Fig 3. *S. haematobium* studies forest plot of mean egg counts and bias table.**

14 (1/6 for *S. mansoni* and 4/8 for *S. haematobium*) of the children aged 6 to <10 years and 2/14 (0/6 for *S. mansoni* and 2/8 for *S. haematobium*) of the children aged 10–14 years).

The centile distribution of the individual-patient ERRs is displayed in Figs 9 and 10 for *S. mansoni* and *S. haematobium*, respectively.

The percentage of patients with ERRs = 0 (no decrease), between >0 and <100%, and 100% (corresponding to the CR) in the different age categories is represented in the bar graphs for each study and in Table 5.

A majority of ERRs are in the 100% category (i.e. 'cured' from the current infection). For preschool-age children as well as younger school-age children (aged 6 to 10 years) almost 80% of the subjects have an ERR above 70%, whereas the results per studies are highly hetegeneous in the 10–14 years old for both *S. mansoni* and *S. haematobium* as ascertained by the very different cumulative curves. There was a significant difference between age groups (p <0.001) for both species, though for different reasons: for *S. mansoni* the difference is driven by preschool-age children having about twice as many non-responders as school-age children (10.2% versus 5.1%), while for *S. haematobium* more preschool-age children were cured (100% ERR: 77.4% versus 60.2%). However, no age difference was seen in treatment outcomes after multivariable adjustment in statistical models. The general linear model of individual-participant ERR with baseline log-transformed egg count as covariate and study, age, and sex as fixed variables did

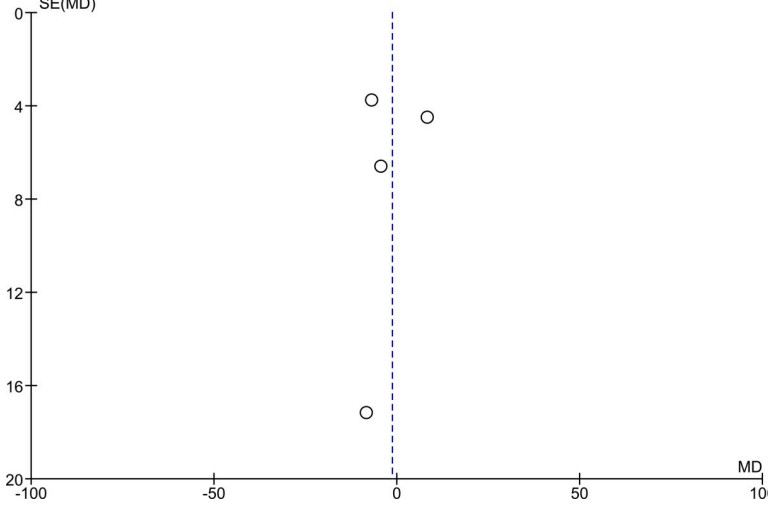

**Fig 4. Studies funnel plot for *S. mansoni*.**

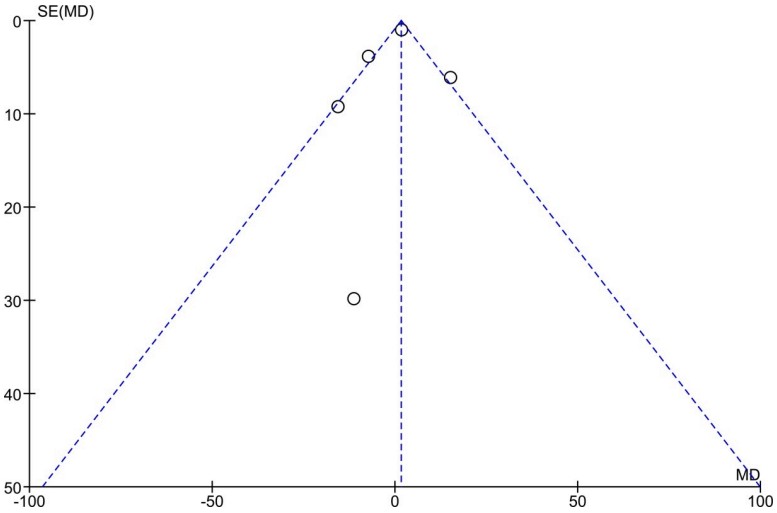

**Fig 5. Studies funnel plot for *S. haematobium*.**

not show any difference in efficacy between age categories in the post-hoc pairwise comparisons of marginal means (least squares means) for either *S. mansoni* or *S. haematobium* (Table 6). This was confirmed in a mixed model employing a random effect for each study (Table 7). Neither baseline egg counts nor duration of follow-up influenced treatment outcome (S7 Table and S8 Table). Sensitivity analyses with log-transformed baseline egg counts for both *S. mansoni* and *S. haematobium* provided similar results for studies accounted for

**Table 3. Number of subjects analyzed (evaluable subjects with follow-up) by age category and *Schistosoma* species.**

| Author, year [Ref.] | *S. mansoni* | | | *S. haematobium* | | | All | | |
|---|---|---|---|---|---|---|---|---|---|
| | 0 to <6 years | 6 to <10 years | 10 to 14 years | 0 to <6 years | 6 to <10 years | 10 to 14 years | 0 to <6 years | 6 to <10 years | 10 to 14 years |
| Coulibaly 2011 [12] | 35 | 0 | 0 | 18 | 0 | 0 | 53 | 0 | 0 |
| Coulibaly 2017 [13] | 38 | 22 | 20 | 0 | 0 | 0 | 38 | 22 | 20 |
| Coulibaly 2018 [14] | 0 | 0 | 0 | 37 | 25 | 12 | 37 | 25 | 12 |
| Garba 2007 [15] | 0 | 99 | 82 | 0 | 211 | 157 | 0 | 310 | 239 |
| Garba 2009 [15] | 0 | 0 | 0 | 0 | 177 | 170 | 0 | 177 | 170 |
| Garba 2013 [16] | 88 | 0 | 0 | 161 | 0 | 0 | 231 | 0 | 0 |
| Garba 1996 [17] | 0 | 0 | 0 | 61 | 85 | 75 | 61 | 85 | 75 |
| Mutapi 2010 [19] | 0 | 0 | 0 | 21 | 115 | 34 | 21 | 115 | 34 |
| Campagne, 2008 [11] | 0 | 0 | 0 | 0 | 30 | 60 | 0 | 30 | 60 |
| Olds, 1999 [21] | 0 | 0 | 0 | 2 | 49 | 67 | 2 | 49 | 67 |
| Olliaro 2007 [22] | 0 | 0 | 190 | 0 | 0 | 0 | 0 | 0 | 190 |
| Raso 2004 [23] | 4 | 12 | 22 | 0 | 0 | 0 | 4 | 12 | 22 |
| Scherrer 2007 [24] | 6 | 22 | 21 | 0 | 0 | 0 | 6 | 22 | 21 |
| Sousa-Figueiredo 2012 [25] | 211 | 94 | 0 | 0 | 0 | 0 | 211 | 94 | 0 |
| Stete 2010 [26] | 0 | 0 | 0 | 1 | 20 | 56 | 1 | 20 | 56 |
| Utzinger 1997 [27] | 0 | 27 | 56 | 0 | 0 | 0 | 0 | 27 | 56 |
| TOTAL | 382 | 276 | 391 | 301 | 712 | 631 | 665 | 988 | 1,022 |
| Preschool-age (0 to <6 years) | 382 | | | 301 | | | 665 | | |
| School-age (6 to 14 years) | | 667 | | | 1,343 | | | 2,010 | |

**Table 4. Intensity of infection at baseline and follow-up, and treatment outcomes expressed as arithmetic mean (AM) egg reduction rate (ERR), cure rate (CR), and mean individual egg reduction rate (all with 95% confidence intervals) by age category and by *Schistosoma* species.**

| Age category | Follow-up duration (in days) | N evaluable | Mean EPG at baseline | Mean EPG at follow-up | ERR 95%CI | CR 95%CI | Mean individual ERR 95%CI |
|---|---|---|---|---|---|---|---|
| *S. mansoni* | | | | | | | |
| 0 to <6 years | 21 | 290 | 244.4 | 51.2 | 79.1 (67.8; 88.6) | 59.7% (54.0%; 65.3%) | 48.8 (23.3; 74.3) |
| 0 to <6 years | 42 | 92 | 109.6 | 7.5 | 93.2 (89.5; 96.3) | 76.1% (67.4%; 84.8%) | 81.1 (70.5; 91.8) |
| 6 to <10 years | 21 | 138 | 226.4 | 23.7 | 89.5 (78.3; 96.3) | 71.7% (64.2%; 79.3%) | 78.2 (60.8; 95.7) |
| 6 to <10 years | 42 | 138 | 100.6 | 26.1 | 74.1 (65.3; 82.1) | 59.4% (51.2%; 67.6%) | 64.1 (43.1; 85.1) |
| 10 to 14 years | 21 | 231 | 21.8 | 0.8 | 96.3 (93.2; 98.8) | 87.9% (83.7%; 92.1%) | 91.6 (84.2; 99.0) |
| 10 to 14 years | 42 | 160 | 114.4 | 15.0 | 86.9 (79.1; 93.3) | 66.3% (58.9%; 73.6%) | 81.9 (72.3; 91.4) |
| 6 to 14 years | 21 | 369 | 98.3 | 9.4 | 90.5 (81.0; 96.3) | 81.8% (77.9%; 85.8%) | 93.5 (91.3; 95.7) |
| 6 to 14 years | 42 | 298 | 108.0 | 20.1 | 81.4 (75.4; 86.4) | 63.1% (57.6%; 68.6%) | 83.4 (79.9; 86.9) |
| *S. haematobium* | | | | | | | |
| 0 to <6 years | 21 | 56 | 20.0 | 0.3 | 98.4 (96.7; 99.5) | 82.1% (72.1%; 92.2%) | 94.9 (89.6; 100.2) |
| 0 to <6 years | 42 | 184 | 37.0 | 5.2 | 85.9 (69.1; 98.8) | 83.7% (78.4%; 89.0%) | 93.9 (90.8; 96.9) |
| 0 to <6 years | 60 | 61 | 14.9 | 22.4 | -50.4 (-147.3; 34.1) | 54.1% (41.6%; 66.6%) | 75.7 (65.5; 86.0) |
| 6 to <10 years | 21 | 222 | 95.0 | 1.9 | 98.1 (97.1; 98.8) | 57.2% (50.7%; 63.7%) | 95.6 (93.7; 97.5) |
| 6 to <10 years | 42 | 405 | 76.7 | 17.0 | 77.9 (67.0; 88.3) | 72.1% (67.7%; 76.5%) | 92.7 (90.5; 94.9) |
| 6 to <10 years | 60 | 85 | 79.1 | 66.5 | 15.9 (-37.2; 56.0) | 22.4% (13.5%; 31.2%) | 59.6 (50.0; 69.2) |
| 10 to 14 years | 21 | 238 | 97.2 | 5.6 | 94.2 (88.9; 98.0) | 56.7% (50.4%; 63.0%) | 93.4 (90.9; 95.8) |
| 10 to 14 years | 42 | 318 | 87.5 | 24.3 | 72.2 (58.1; 83.9) | 67.9% (62.8%; 73.1%) | 91.3 (88.8; 93.9) |
| 10 to 14 years | 60 | 75 | 81.4 | 35.1 | 57.0 (37.9; 73.1) | 25.3% (15.5%; 35.2%) | 61.5 (52.2; 70.9) |
| 6 to 14 years | 21 | 460 | 96.1 | 3.8 | 96.0 (93.1; 98.1) | 57.0% (52.4%; 61.5%) | 94.4 (92.9; 96.0) |
| 6 to 14 years | 42 | 723 | 81.5 | 20.2 | 75.2 (67.6; 82.8) | 70.3% (66.9%; 73.6%) | 92.1 (90.5; 93.7) |
| 6 to 14 years | 60 | 160 | 80.2 | 51.7 | 35.5 (9.1; 57.3) | 23.8% (17.2%; 30.3%) | 60.5 (53.8; 67.2) |

either as fixed factor (general linear model, S4 Table) or as random effect (mixed model, S5 Table).

Safety was reported in 7/16 studies with only four studies [13, 16, 19, 35] reporting on evaluable patients who had safety data corresponding to participants with follow-up <60 days, age ≤14 years, and praziquantel dose of 40 mg/kg (total number assessed on day 1 = 1,128; at follow-up = 1,065, 94%) (Table 8). Overall, 226 (20%) patients suffered from at least one AE 4 hours after drug intake, 88 (8%) after 24 hours, and 33 (3%) at the treatment follow-up,

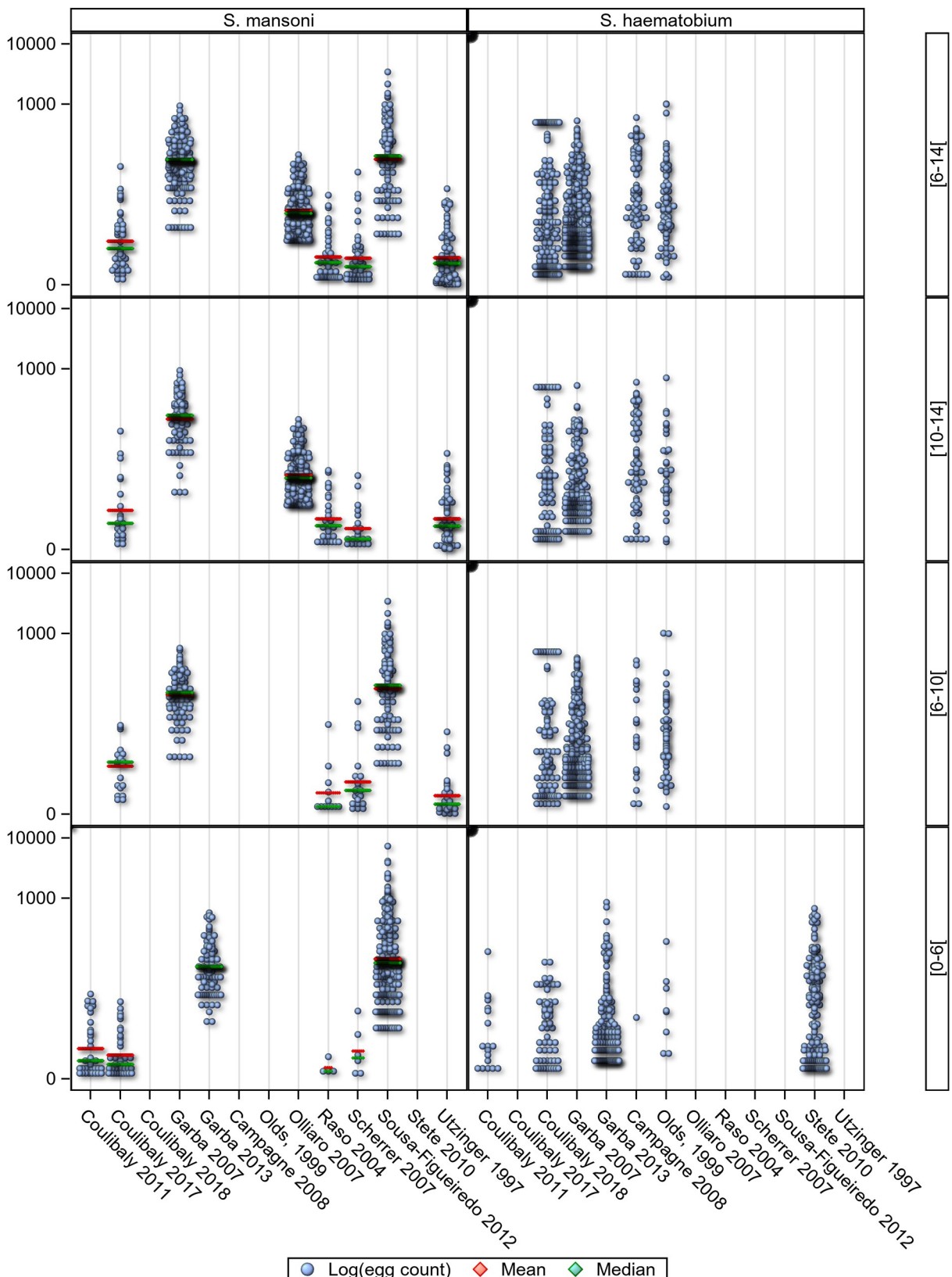

**Fig 6. Baseline intensity of infection analyses adjusted for study and age.**

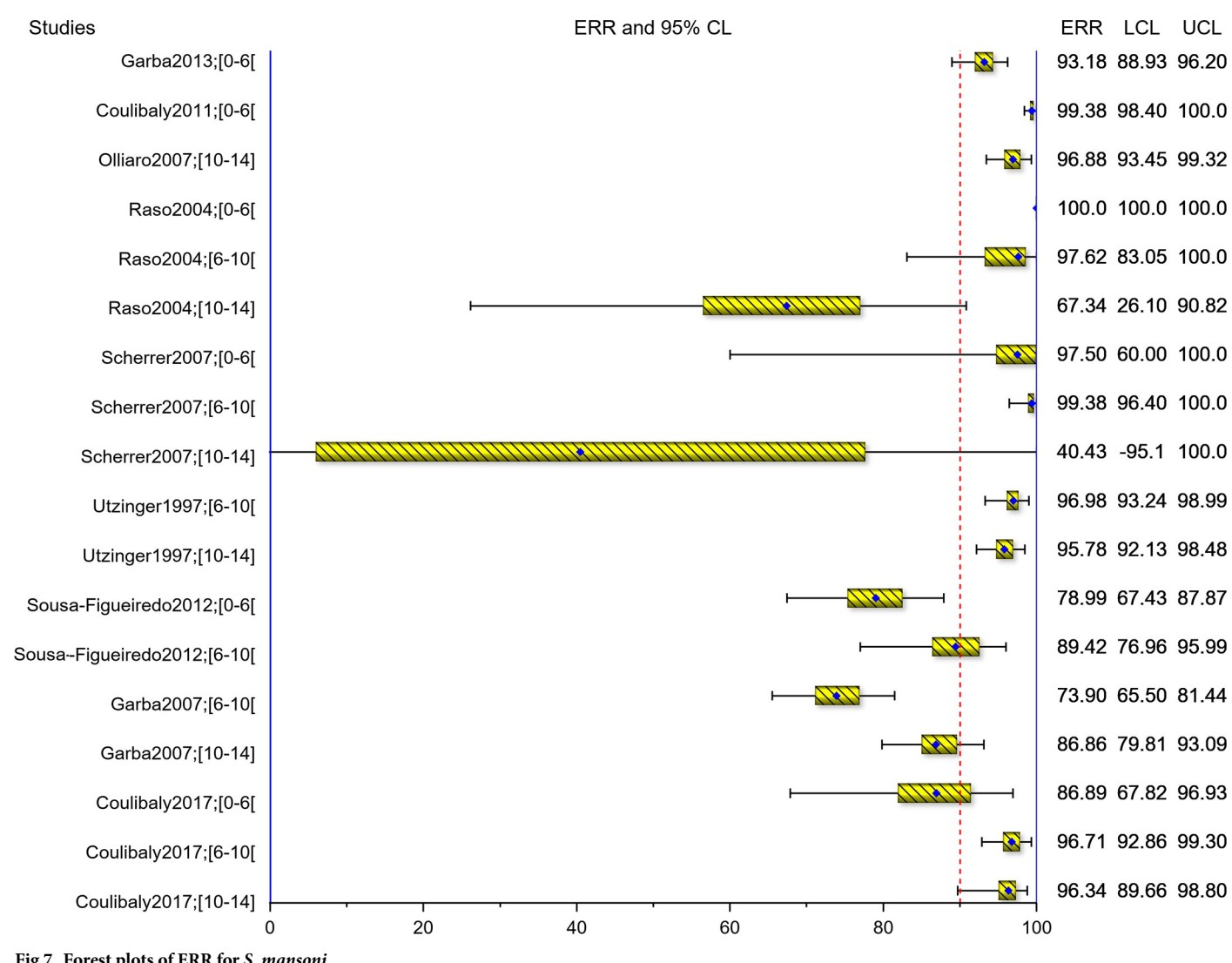

**Fig 7. Forest plots of ERR for *S. mansoni*.**

respectively. At least one AE was experienced at 4 hours post-treatment by 19% and 44% of under 6-year-old children and children aged 10–14 years, respectively. Only one child had an AE in the 6- to <10-year-old age group. The relative proportions at 24 hours were 5%, 21%, and 0%. At follow-up, 10% of the children aged 10–14 years reported at least one AE, and none in the other age-groups.

A total of 370, 272, and 134 AEs were reported on the day of treatment 4 hours after drug intake, at 24 hours post-treatment, and at the follow-up visit, respectively. At 4 hours, there were 81 AEs of mild intensity in children aged 0 to <6 years; one mild in the age group 6 to <10 years; and 228 mild, 159 moderate, and one severe AEs in children aged 10–14 years. At 24 hours, 19 mild AEs in the 0 to <6 year age group; none in the children aged 6 to <10 years; and 205 mild, 46 moderate, and two severe AEs in the oldest group of children 10–14 years. At treatment follow-up, a total of 88 mild and 46 moderate AEs were observed in the 10- to 14-year-old age group. The severe AEs were dizziness, anorexia, and diarrhea, but none of the AEs were judged as serious.

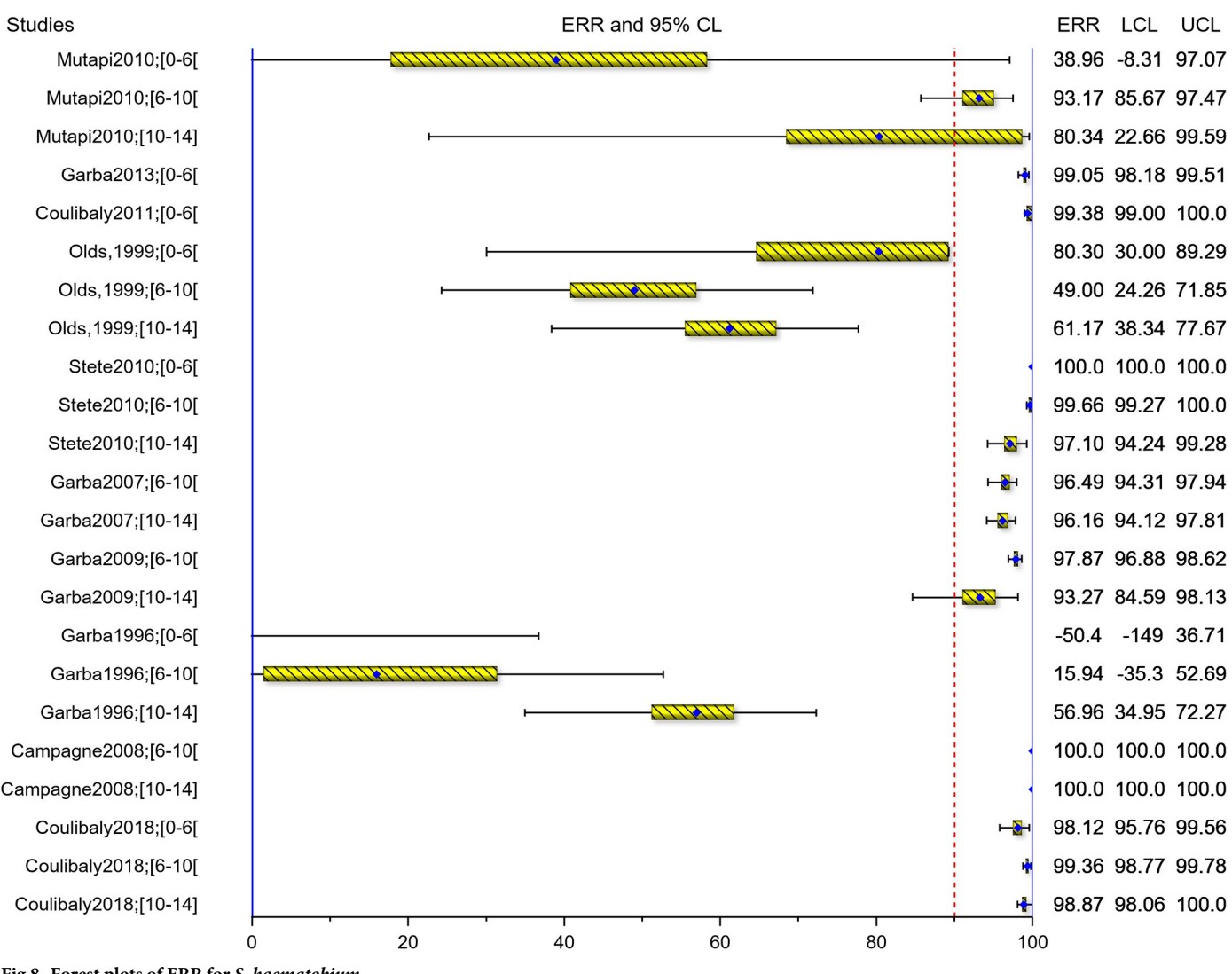

| Studies | ERR | LCL | UCL |
|---|---|---|---|
| Mutapi2010;[0-6[ | 38.96 | -8.31 | 97.07 |
| Mutapi2010;[6-10[ | 93.17 | 85.67 | 97.47 |
| Mutapi2010;[10-14] | 80.34 | 22.66 | 99.59 |
| Garba2013;[0-6[ | 99.05 | 98.18 | 99.51 |
| Coulibaly2011;[0-6[ | 99.38 | 99.00 | 100.0 |
| Olds,1999;[0-6[ | 80.30 | 30.00 | 89.29 |
| Olds,1999;[6-10[ | 49.00 | 24.26 | 71.85 |
| Olds,1999;[10-14] | 61.17 | 38.34 | 77.67 |
| Stete2010;[0-6[ | 100.0 | 100.0 | 100.0 |
| Stete2010;[6-10[ | 99.66 | 99.27 | 100.0 |
| Stete2010;[10-14] | 97.10 | 94.24 | 99.28 |
| Garba2007;[6-10[ | 96.49 | 94.31 | 97.94 |
| Garba2007;[10-14] | 96.16 | 94.12 | 97.81 |
| Garba2009;[6-10[ | 97.87 | 96.88 | 98.62 |
| Garba2009;[10-14] | 93.27 | 84.59 | 98.13 |
| Garba1996;[0-6[ | -50.4 | -149 | 36.71 |
| Garba1996;[6-10[ | 15.94 | -35.3 | 52.69 |
| Garba1996;[10-14] | 56.96 | 34.95 | 72.27 |
| Campagne2008;[6-10[ | 100.0 | 100.0 | 100.0 |
| Campagne2008;[10-14] | 100.0 | 100.0 | 100.0 |
| Coulibaly2018;[0-6[ | 98.12 | 95.76 | 99.56 |
| Coulibaly2018;[6-10[ | 99.36 | 98.77 | 99.78 |
| Coulibaly2018;[10-14] | 98.87 | 98.06 | 100.0 |

**Fig 8. Forest plots of ERR for *S. haematobium*.**

At 4 hours post-treatment, the most frequent AE was abdominal pain (39.7% of AEs), followed by diarrhea (14.3%), vomiting (9.5%), headache and dizziness (both 8.4%). At 24 hours, it was headache (23.2%) then abdominal pain (16.5%), dizziness (15.1%), and diarrhea (11%). At treatment follow-up, headache (27.6%), abdominal pain (19.4%), anorexia (14.2%), and dizziness (12.7%) were the most frequent AEs.

## Discussion

This individual-participant data meta-analysis aimed to assess whether a single 40 mg/kg oral dose of 40 mg/kg of praziquantel administered to preschool-age children (aged <6 years) would be as efficacious and safe as in school-age children. This kind of evidence is important. Indeed, while the current emphasis of preventive chemotherapy is on school-age children [1, 34], preschool-age children are now also recognized to carry a significant burden of disease, and hence, they are becoming a target for preventive chemotherapy, especially if a pediatric formulation of praziquantel becomes available [6, 7]. Furthermore, efforts are under way to strategically move from morbidity control to elimination, which means preventive

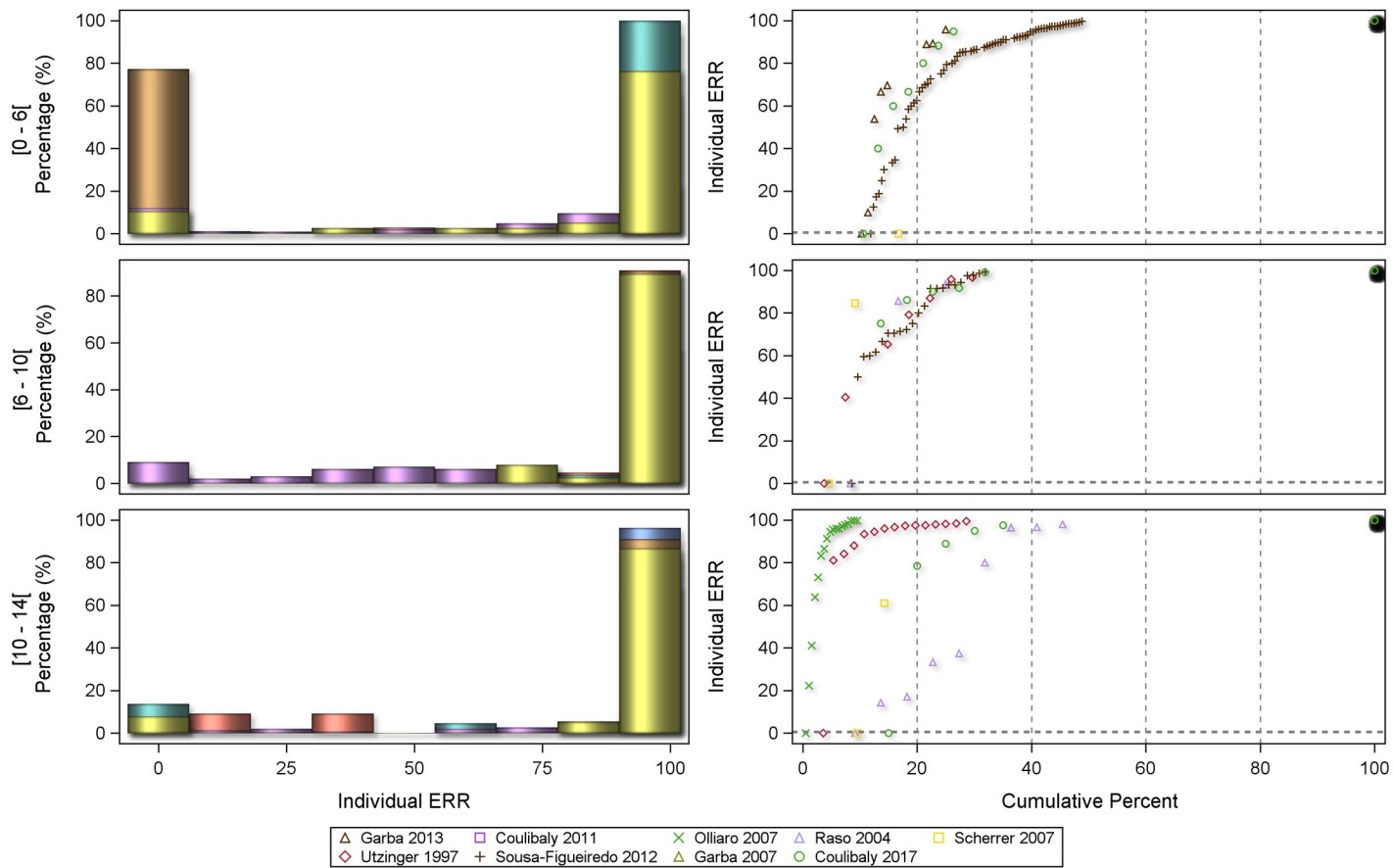

**Fig 9. Centile distribution of the individual-patient ERRs for *S. mansoni*.**

chemotherapy might need to be expanded to include all age-groups in order to remove all untreated reservoirs of infection that contribute to ongoing local transmission [40].

Taken together, our results point to no age-effect on treatment efficacy with the standard single-dose of 40 mg/kg praziquantel, whether administered to children under the age of 6 years (preschool-age), or to school-age children (aged 6–14 years), or whether the latter group is further broken down into 6 to less than 10, and 10 through 14 years (as some studies only included either age-group).

We analyzed data using two models: (i) a general linear model with or without baseline log-transformed *Schistosoma* egg counts as covariate and study, age, and sex as cofactors; and (ii) a mixed model with or without baseline log-transformed egg counts as covariate, age as a fixed factor, with a random effect for each study. None of these models detected a statistically significant difference in the individual ERRs between age categories. These analyses also show that the lower baseline egg counts found in preschool-age children did not have an effect on treatment outcome, nor did duration of follow-up within the 21–60 day time-frame.

Our conclusions are further supported by the observation that, when using the WHO-recommended AM-ERR, there is no indication that fewer groups composed of preschool-age children meet the WHO ≥95% efficacy threshold compared to school-age children. These results are in overall agreement with mixed-effect multivariate analysis of aggregated data that found no significant difference between preschool- and school-age children for CR or geometric mean ERR for either *S. mansoni* or *S. haematobium* after controlling for time of assessment,

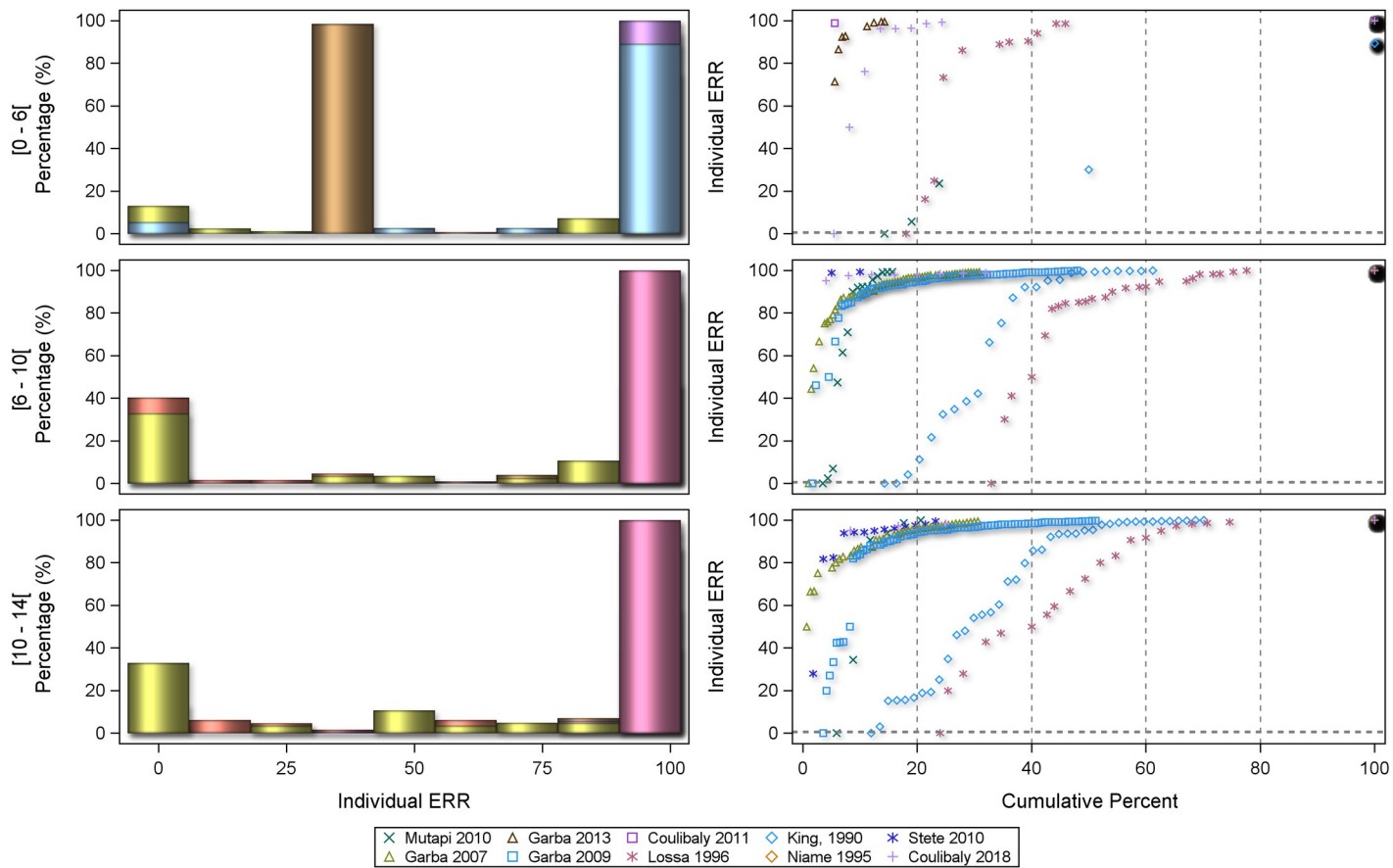

**Fig 10. Centile distribution of the individual-patient ERRs for *S. haematobium*.**

formulation, intensity of infection, and diagnostic approach [41]. While host factors, like age-related differences in drug metabolizing activity [42] might play a role in praziquantel metabolism and overall efficacy, increasing the dose to 60 mg/kg may not lead to better efficacy. Two of the studies conducted in Côte d'Ivoire contributing to this analysis, which also studied the response to placebo and doses ranging from 20 mg/kg to 60 mg/kg, showed a flat dose-response curve in both preschool- and school-age children infected with *S. haematobium* and preschool-children infected with *S. mansoni* [13, 14]. Another study [22] compared 40 mg/kg to 60 mg/kg in children aged 10–14 years, and did not find a difference either. Similarly, in an aggregated-data meta-analysis using a random-effect meta-analysis regression model, a dose-effect for CR was found up to 40 mg/kg for *S. mansoni* and 30 mg/kg for *S. haematobium* with no benefit in increasing the dose, in school-age children [43].

Treatment with a single 40 mg/kg oral dose of praziquantel was well tolerated, especially in preschool-age children who experienced fewer and generally only mild AEs when compared to school-age children. Previous systematic reviews pertaining to the efficacy and safety of praziquantel for schistosomiasis already showed that praziquantel is safe and AEs usually mild and self-limiting [41, 44, 45]. It is, however, conceivable that this is a result of underreporting by younger children, if AEs were not specifically and proactively elicited.

Our study has several limitations. First, out of 16 studies included in the analysis, 13 were single-arm thus with an unclear risk of bias concerning selection of participants. Second, the studies enrolled selectively different age groups, so we used the principles of network meta-

**Table 5. Individual patient egg reduction rate (ERR) by *Schistosoma* species and age group.**

| Age | ERR | *S. mansoni* | | *S. haematobium* | |
|---|---|---|---|---|---|
| | | n | % | n | % |
| 0 to <6 years | 0% | 39 | 10.2 | 19 | 6.3 |
| | 0<ERR<100 | 100 | 26.2 | 49 | 16.3 |
| | 100% | 243 | 63.6 | 233 | 77.4 |
| 6 to <10 years | 0% | 20 | 7.3 | 44 | 6.2 |
| | 0<ERR<100 | 75 | 27.2 | 230 | 32.3 |
| | 100% | 181 | 65.6 | 438 | 61.5 |
| 10 to 14 years | 0% | 14 | 3.6 | 34 | 5.4 |
| | 0<ERR<100 | 68 | 17.4 | 227 | 36.0 |
| | 100% | 309 | 79.0 | 370 | 58.6 |
| $X^2$ p-value | | | <0.001 | | <0.001 |
| 6 to 14 years | 0% | 34 | 5.1 | 78 | 5.8 |
| | 0<ERR<100 | 143 | 21.4 | 457 | 34.0 |
| | 100% | 490 | 73.5 | 808 | 60.2 |
| $X^2$ p-value | | | <0.001 | | <0.001 |

analysis to compare participant outcomes in different groups, which were not necessarily enrolled at the same site and time, and might thus not be comparable. Third, safety and tolerability at large were also generally underreported. Fourth, no information is available for S. *japonicum*. A more general methodological issue is the limitations of counting eggs in excreta in young children, and the imperfect correlation between egg shedding and worm burden, which would be better reflected by antigen-detection methods. [46] However, the significance and comparability of results of efficacy studies based on antigen-detection methods vis-à-vis direct egg-detection remains unclear.

In conclusion, based on the data presented here, there is no indication that preventive chemotherapy with single-dose 40 mg/kg praziquantel would be less efficacious and tolerated in preschool-age children than it is in school-age children. However, these conclusions are drawn from an available, limited body of evidence, which is only a tiny fraction of the overall use of praziquantel, with only a few studies designed for direct comparisons between age groups. Whether and when preschool-age children will be exposed systematically to preventive chemotherapy, close monitoring of efficacy as well as safety will be required. This in particular

**Table 6. Post-hoc pairwise comparisons between age categories, general linear model.**

| Effect | Category | | *S. mansoni* individual egg reduction rates | | | *S. haematobium* individual egg reduction rates | | |
|---|---|---|---|---|---|---|---|---|
| | | | Mean | SD | Adj P | Mean | SD | Adj P |
| Age | 10 to 14 years | | 0.937 | 0.0224 | | 0.892 | 0.0134 | |
| | 6 to <10 years | | 0.925 | 0.0205 | | 0.903 | 0.0130 | |
| | 0 to <6 years | | 0.884 | 0.0238 | | 0.929 | 0.0203 | |
| Age | 6 to 14 years | | 0.930 | 0.0162 | | 0.898 | 0.0113 | |
| | 0 to <6 years | | 0.887 | 0.0223 | | 0.928 | 0.0202 | |
| Pairwise comparisons | | | | | | | | |
| Age | 10 to 14 years | 6 to <10 years | 0.0119 | 0.0280 | 0.906 | -0.0106 | 0.0134 | 0.710 |
| | 10 to 14 years | 0 to <6 years | 0.0533 | 0.0372 | 0.324 | -0.0368 | 0.0273 | 0.368 |
| | 6 to <10 years | 0 to <6 year | 0.0414 | 0.0288 | 0.322 | -0.0262 | 0.0266 | 0.585 |
| Age | 6 to 14 years | 0 to <6 years | 0.0432 | 0.0285 | 0.130 | -0.0304 | 0.0260 | 0.243 |

**Table 7.  Post-hoc pairwise comparisons between age categories, mixed model.**

| Effect | Category | | S. mansoni individual egg reduction rates | | | S. haematobium individual egg reduction rates | | |
|---|---|---|---|---|---|---|---|---|
| | | | Mean | SD | Adj P | Mean | SD | Adj P |
| Age | 10 to 14 years | | 0.931 | 0.03744 | | 0.8902 | 0.04094 | |
| | 6 to <10 years | | 0.913 | 0.03679 | | 0.9006 | 0.04083 | |
| | 0 to <6 yeras | | 0.875 | 0.03796 | | 0.9292 | 0.04367 | |
| Age | 6 to 14 years | | 0.922 | 0.03501 | | 0.8957 | 0.0403 | |
| | 0 to <6 years | | 0.879 | 0.03779 | | 0.9284 | 0.04363 | |
| Pairwise comparisons | | | | | | | | |
| Age | 10 to 14 years | 6 to <10 years | 0.0172 | 0.0272 | 0.802 | -0.0105 | 0.0134 | 0.715 |
| | 10 to 14 years | 0 to <6 years | 0.0558 | 0.0344 | 0.236 | -0.0390 | 0.0265 | 0.306 |
| | 6 to <10 years | 0 to <6 years | 0.0386 | 0.0275 | 0.341 | -0.0285 | 0.0259 | 0.512 |
| Age | 6 to 14 years | 0 to <6 years | 0.0423 | 0.0269 | 0.117 | -0.0327 | 0.0253 | 0.170 |

**Table 8.  Number of children treated with a single 40 mg/kg oral dose of praziquantel reporting adverse events (AEs) and number of AEs by age group.**

| | | 0 to <6 years | | 6 to <10 years | | 10 to 14 years | | All | |
|---|---|---|---|---|---|---|---|---|---|
| | | N | % | N | % | N | % | N | % |
| Day 0 (4 hours post-treatment) | Experienced ≥1 AE | 81 | 19% | 1 | 0% | 144 | 44% | 226 | 20% |
| | Number of AEs | 81 | 22% | 1 | 0% | 288 | 78% | 370 | |
| | Number evaluated | 416 | | 381 | | 331 | | 1,128 | |
| Day 1 (24 hours post-treatment) | Experienced ≥1 AE | 19 | 5% | | | 69 | 21% | 88 | 8% |
| | Number of AEs | 19 | 7% | | | 253 | 93% | 272 | |
| | Number evaluated | 416 | | 381 | | 331 | | 1128 | |
| Day 21 | Experienced ≥1 AE | | | | | 33 | 10% | 33 | 3% |
| | Number of AEs | | | | | 134 | 1 | 134 | |
| | Number evaluated | 372 | | 376 | | 317 | | 1070 | |

considering preschool-age children have about twice as high rates of non-responders in the individual-patient ERR analysis than older children.

It should also be noted that the WHO-recommended 95% ERR threshold was not met in many studies, which calls for improved dosing and delivery strategies.

## Supporting information

**S1 Table. Species and diagnostic approach used (excluded studies shaded).**
(DOCX)

**S2 Table. Intensity of infection and treatment outcomes by study for *Schistosoma mansoni*.**
(DOCX)

**S3 Table. Intensity of infection and treatment outcomes by study for *Schistosoma haematobium*.**
(DOCX)

**S4 Table. Results of general linear model of individual ERR with baseline log transformed egg count as covariate and studies, age and sex as factors.**
(DOCX)

**S5 Table. Results of mixed model of individual ERR with baseline log transformed egg count as covariate, age and sex as factors, and with random effect for the study.**
(DOCX)

**S6 Table.** Results of general linear model of individual a) *S. mansoni* and b) *S. haematobium* baseline log transformed egg count with study, age, and sex as factors.
(DOCX)

**S7 Table.** Results of general linear model of individual *S. mansoni* ERR without baseline log transformed egg count as covariate and studies and age as factors (a: 3 categories, b: 2 categories).
(DOCX)

**S8 Table.** Results of mixed model of individual *S. haematobium* ERR without baseline log transformed egg count as covariate and age as a fixed factor, with studies as a random factor (a: 3 categories, b: 2 categories).
(DOCX)

**S9 Table. PRISMA checklist.**
(DOCX)

**S10 Table. PRISMA flow diagram.**
(DOCX)

**S1 Fig. Network of possible preschool- and school-age children comparisons across all studies.**
(TIF)

**S2 Fig. Age distribution of participants by Schistosoma species.**
(TIF)

## Acknowledgments

The authors would like to express their big thanks to the authors of the original publications used in this work for having kindly provided the individual participants data.

## Disclaimer

AG and CH are staff members of the World Health Organization, FM is a staff member of NIHR and MV is a staff member of the Luxembourg Institute of Health; the authors alone are responsible for the views expressed in this publication and it does not necessarily represent the decisions, policy, or views of their respective organizations.

## Author Contributions

**Conceptualization:** Piero L. Olliaro, Michel T. Vaillant.

**Data curation:** Michel T. Vaillant.

**Formal analysis:** Michel T. Vaillant.

**Investigation:** Piero L. Olliaro, Jean T. Coulibaly, Amadou Garba, Jennifer Keiser, Charles H. King, Francisca Mutapi, Eliézer K. N'Goran, Giovanna Raso, Alexandra U. Scherrer, José Carlos Sousa-Figueiredo, Katarina Stete, Jürg Utzinger.

**Methodology:** Piero L. Olliaro, Michel T. Vaillant.

**Project administration:** Piero L. Olliaro, Christine Halleux.

**Resources:** Piero L. Olliaro, Jennifer Keiser, Charles H. King, Francisca Mutapi, Eliézer K. N'Goran, Giovanna Raso, Alexandra U. Scherrer, José Carlos Sousa-Figueiredo, Katarina Stete, Jürg Utzinger.

**Software:** Michel T. Vaillant.

**Supervision:** Piero L. Olliaro.

**Validation:** Jürg Utzinger, Michel T. Vaillant.

**Writing – original draft:** Piero L. Olliaro, Michel T. Vaillant.

**Writing – review & editing:** Piero L. Olliaro, Jean T. Coulibaly, Amadou Garba, Christine Halleux, Jennifer Keiser, Charles H. King, Francisca Mutapi, Eliézer K. N'Goran, Giovanna Raso, Alexandra U. Scherrer, José Carlos Sousa-Figueiredo, Katarina Stete, Jürg Utzinger, Michel T. Vaillant.

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
