## [Decision Letter · Decision Letter 0]

5 Mar 2020

Dear Vaillant,

Thank you very much for submitting your manuscript "Efficacy and safety of single 40 mg/kg oral praziquantel in the treatment of schistosomiasis in preschool-age versus school-age children: an individual participant data meta-analysis" for consideration at PLOS Neglected Tropical Diseases. As with all papers reviewed by the journal, your manuscript was reviewed by members of the editorial board and by several independent reviewers. The reviewers appreciated the attention to an important topic. Based on the reviews, we are likely to accept this manuscript for publication, providing that you modify the manuscript according to the review recommendations. 

Sincerely,

Antonio Montresor

Guest Editor

Timothy Geary

Deputy Editor

Reviewer's Responses to Questions

**Key Review Criteria Required for Acceptance?**

**Methods**

-Are the objectives of the study clearly articulated with a clear testable hypothesis stated?

-Is the study design appropriate to address the stated objectives?

-Is the population clearly described and appropriate for the hypothesis being tested?

-Is the sample size sufficient to ensure adequate power to address the hypothesis being tested?

-Were correct statistical analysis used to support conclusions?

-Are there concerns about ethical or regulatory requirements being met?

Reviewer #1: Did all studies deliver praziquantel to the PSACs in the same way - whole tablet, crushed tablet in juice? 

Were all treatments accompanied by food ? If not, did this have an impact on adverse events?

Was the praziquantel used in the studies all from the same source / manufacturer?

I see little value including S. japonicum due to the stated absence of data.

Reviewer #2: A very clear statistical analysis from a well-organised database of IPD.

Reviewer #3: General

1) Were PRISMA guidelines used for this Meta-analysis. Not referenced in methods. You provide the flow diagram but should reference in methods. 

2) Regarding S. japonicum, there is only ONE PSAC subjects across all studies. One PSAC in China, none in Philippines. How is this useful for meta with focus on treatment safety and efficacy in PSAC??? At the very least references to S. japonicum should be removed from abstract and other places. 

3) in many tables the close brackets [ ] is left open [ [ for many age groups

5) In methods, please clarify what was asked regarding AEs at follow up. Did you ask about current AE due to the (weeks preceding) treatment or recall of AE in day or so after?

6) Methods do not approach whether studies referenced repeating dose if vomiting occurred or rates of refusal among young children. Although some studies likely didn’t collect, one can still add to discussion regarding ways to improve overall ERR for all children. See point 6 above. Lines 153-55. It is not clear here whether you are referring to reporting bias (participant attribute) or publication bias. These lines seem to apply to the latter. 

Specific

Line 164-spell out AM-ERR with first use 

Line 233- as worded it is not clear how you are supporting lack of different types of bias:

“The RCTs were at low risk of selection bias with computer-generated block randomization, adequate allocation concealment, and blinding of either participants, personnel, or outcome assessment.”

Some of these procedures do not support “selection bias”

**Results**

-Does the analysis presented match the analysis plan?

-Are the results clearly and completely presented?

-Are the figures (Tables, Images) of sufficient quality for clarity?

Reviewer #1: OK

Reviewer #2: Some excellent graphs showing novelty in visualization of complex data.

Reviewer #3: See general comments on S. japonicum

Lines 310-12 

This sentence is not clear:

At least one AE was experienced at 4 hours post-treatment by 19% and 44% of under 6-year-old children and children aged 10-14 years, respectively.

Table 6

Please provide clearer title and column headings to denote what the “mean” represents

Figure 6 – need to clarify what different colors represent in the bar graph. Large number of low ERR in one study? As above, this also needs to be discussed. Neither the figure legend nor results text explain this figure at all adequately. 

In many tables the close brackets [ ] is left open [ [ for many age groups

**Conclusions**

-Are the conclusions supported by the data presented?

-Are the limitations of analysis clearly described?

-Do the authors discuss how these data can be helpful to advance our understanding of the topic under study?

-Is public health relevance addressed?

Reviewer #1: OK

Reviewer #2: Large true but I think the authors should elaborate a bit more on the following:

Egg count is not the 'best' measure of cure in young children (since there is an unusual dynamic across time of first infection and age towards patency (see https://www.ncbi.nlm.nih.gov/pubmed/21245910). I realize that egg-count is important in SAC dynamics but some awareness of this non-linear effect in PSAC is important.

By that same token a bit more should be made of antigen biomarkers (either CAA in sera/urine or CCA in urine). 

Collectively I expect the use of biomarkers would show less efficacy, and note that the paediatric praziquantel consortium (on the basis of their trials) is adopting 50 mg/kg of the L-PZQ (given inferiority/superiority measures).

The future adoption of a mono-isomeric form in PSAC somewhat changes the present guidance that this current script offers. It does support however the WHO-stop gap of using crushed racemate in this age class. From the results presented here seems to be a sensible altrnative way forward more immediate to the L-PZQ (which might need testing in SAC?).

Reviewer #3: Lines 340-1- same issue on emphasis on formulation. Most of your included studies used crushed tablets

Lines 365- 

Discussion focuses too heavily on lack of difference in other studies at 60 mg/kg. Though you show that the ERR not significantly different between age groups at 40 mg/kg, there is not discussion of the fact that most studies did not find ERR at or above WHO recommended 95% threshold. This must be discussed. There are other reasons besides dose provided that may explain including different dosing strategies (larger total dose over more time to improve tolerability, formulations that might improve drug delivery based on tolerability, absorption etc.

Also, at some point in the discussion, particularly when you reference issue of higher doses to 60 mg/kg, must state that much less is known about this in S. japonicum. 

Here or elsewhere, must discuss the almost double rate of ERR less than 0% in S. mansoni among PSAC. This can be included in above regarding adequacy of ANY current does. 

Lines379+ - must include in limitations that you cannot say anything about comparing efficacy of 40 mg/kg among PSAC and school age for S. japonicum. That gets lost entirely.

**Editorial and Data Presentation Modifications?**

Reviewer #1: Apart from the comments above "Accept"

Reviewer #2: NA

Reviewer #3: Methods

Lines 153-55. It is not clear here whether you are referring to reporting bias (participant attribute) or publication bias. These lines seem to apply to the latter. 

Line 164-spell out AM-ERR with first use 

Line 233- as worded it is not clear how you are supporting lack of different types of bias:

“The RCTs were at low risk of selection bias with computer-generated block randomization, adequate allocation concealment, and blinding of either participants, personnel, or outcome assessment.”

Some of these procedures do not support “selection bias”

Results

Lines 310-12 

This sentence is not clear:

At least one AE was experienced at 4 hours post-treatment by 19% and 44% of under 6-year-old children and children aged 10-14 years, respectively.

Discussion

Lines 340-1- same issue on emphasis on formulation. Most of your included studies used crushed tablets

Lines 365- 

Discussion focuses too heavily on lack of difference in other studies at 60 mg/kg. Though you show that the ERR not significantly different between age groups at 40 mg/kg, there is not discussion of the fact that most studies did not find ERR at or above WHO recommended 95% threshold. This must be discussed. There are other reasons besides dose provided that may explain including different dosing strategies (larger total dose over more time to improve tolerability, formulations that might improve drug delivery based on tolerability, absorption etc.

Also, at some point in the discussion, particularly when you reference issue of higher doses to 60 mg/kg, must state that much less is known about this in S. japonicum. 

Here or elsewhere, must discuss the almost double rate of ERR less than 0% in S. mansoni among PSAC. This can be included in above regarding adequacy of ANY current does. 

Lines379+ - must include in limitations that you cannot say anything about comparing efficacy of 40 mg/kg among PSAC and school age for S. japonicum. That gets lost entirely.

**Summary and General Comments**

Reviewer #1: The limitations of the study have been addressed by the authors. This is a most useful analysis of data considered in previously published papers and highlights that treatment of PSACs with praziquantel has a similar benefits to treatment of older children.

Reviewer #2: A useful script - secondary analysis - competently done. For brevity I think it could be shortened a little will a few more specific pointers on 'grey' areas (mono-isomer v racemate) where future impact might be less than expected.

Reviewer #3: 1) Were PRISMA guidelines used for this Meta-analysis. Not referenced in methods. You provide the flow diagram but should reference in methods. 

2) Regarding S. japonicum, there is only ONE PSAC subjects across all studies. One PSAC in China, none in Philippines. How is this useful for meta with focus on treatment safety and efficacy in PSAC??? At the very least references to S. japonicum should be removed from abstract and other places. 

3) introduction Lines 97-101: Same issue on waiting for other formulation. You cite the WHO report of 2010 on this which actually states in the executive summary: 

“Studies on the treatment of preschool-age children conducted in these five countries (n=3198) among children aged 1 month to 7 years showed that praziquantel in a tablet or suspension formulation was safe and effective against schistosomiasis, and acceptable…The two studies in Mali and Uganda that compared suspension and tablets found no difference in cure rates between the two formulations. In Uganda, there was also no difference in rates of egg reduction between the two formulations.”

This is not simply an issue of semantics; the way this manuscript reads strongly implies that we must await pediatric formulations which will do harm to young children who can take crushed tablets. Even your PICOS statement addresses efficacy of 40 mg/kg dose as the primary goal of the study, without reference to dosing form which dominates abstract and introduction. 

4) Introduction Lines 99-101- at least one drug (Bayer) is “registered” (US FDA approved) down to age one so this should be modified accordingly. 

4) In methods, please clarify what was asked regarding AEs at follow up. Did you ask about current AE due to the (weeks preceding) treatment or recall of AE in day or so after?

5) at some point in the discussion, particularly when you reference issue of higher doses to 60 mg/kg, must state that much less is known about this in S. japonicum. 

6) Discussion focuses too heavily on lack of difference in other studies at 60 mg/kg. Though you show that the ERR not significantly different between age groups at 40 mg/kg, there is not discussion of the fact that most studies did not find ERR at or above WHO recommended 95% threshold. This must be discussed. There are other reasons besides dose provided that may explain including different dosing strategies (larger total dose over more time to improve tolerability, formulations that might improve drug delivery based on tolerability, absorption etc. 

7) if any authors receive funding from pharmaceutical companies developing Pediatric formulations for Praziquantel, this should be disclosed. 

8) for S. mansoni, there were double the number of children who had ERR 0%. This is never discussed and provides good topic for discussion under tolerability issues in this age group. Did they vomit the dose/refuse? 

9) Methods do not approach whether studies referenced repeating dose if vomiting occurred or rates of refusal among young children. Although some studies likely didn’t collect, one can still add to discussion regarding ways to improve overall ERR for all children. See point 6 above.

PLOS authors have the option to publish the peer review history of their article (what does this mean?). If published, this will include your full peer review and any attached files.

Reviewer #1: No

Reviewer #2: No

Reviewer #3: No
---

## [Editor Report · Decision Letter 1]

8 Apr 2020

Dear Vaillant,

We are pleased to inform you that your manuscript 'Efficacy and safety of single 40 mg/kg oral praziquantel in the treatment of schistosomiasis in preschool-age versus school-age children: an individual participant data meta-analysis' has been provisionally accepted for publication in PLOS Neglected Tropical Diseases.

Best regards,

Antonio Montresor

Guest Editor

Timothy Geary

Deputy Editor

Thank you for addressing the points raised by the reviewers,

the manuscript is in my opinion suitable for publication

---

## [Editor Report · Acceptance letter]

12 Jun 2020

Dear Vaillant,

We are delighted to inform you that your manuscript, "Efficacy and safety of single 40 mg/kg oral praziquantel in the treatment of schistosomiasis in preschool-age versus school-age children: an individual participant data meta-analysis," has been formally accepted for publication in PLOS Neglected Tropical Diseases.

Best regards,

Shaden Kamhawi

co-Editor-in-Chief

Paul Brindley

co-Editor-in-Chief
